# Puzzle out Machine Learning Model-Explaining Disintegration Process in ODTs

**DOI:** 10.3390/pharmaceutics14040859

**Published:** 2022-04-13

**Authors:** Jakub Szlęk, Mohammad Hassan Khalid, Adam Pacławski, Natalia Czub, Aleksander Mendyk

**Affiliations:** Department of Pharmaceutical Technology and Biopharmaceutics, Jagiellonian University Medical College, Medyczna 9, 30-688 Kraków, Poland; j.szlek@uj.edu.pl (J.S.); hassankhalid101@gmail.com (M.H.K.); adam.paclawski@uj.edu.pl (A.P.); natalia.czub@doctoral.uj.edu.pl (N.C.)

**Keywords:** ODTs, machine learning, AutoML, shapley values, partial dependence plots, explainable models, orally disintegrating tablets

## Abstract

Tablets are the most common dosage form of pharmaceutical products. While tablets represent the majority of marketed pharmaceutical products, there remain a significant number of patients who find it difficult to swallow conventional tablets. Such difficulties lead to reduced patient compliance. Orally disintegrating tablets (ODT), sometimes called oral dispersible tablets, are the dosage form of choice for patients with swallowing difficulties. ODTs are defined as a solid dosage form for rapid disintegration prior to swallowing. The disintegration time, therefore, is one of the most important and optimizable critical quality attributes (CQAs) for ODTs. Current strategies to optimize ODT disintegration times are based on a conventional trial-and-error method whereby a small number of samples are used as proxies for the compliance of whole batches. We present an alternative machine learning approach to optimize the disintegration time based on a wide variety of machine learning (ML) models through the H2O AutoML platform. ML models are presented with inputs from a database originally presented by Han et al., which was enhanced and curated to include chemical descriptors representing active pharmaceutical ingredient (API) characteristics. A deep learning model with a 10-fold cross-validation NRMSE of 8.1% and an R^2^ of 0.84 was obtained. The critical parameters influencing the disintegration of the directly compressed ODTs were ascertained using the SHAP method to explain ML model predictions. A reusable, open-source tool, the ODT calculator, is now available at Heroku platform.

## 1. Introduction

Orally disintegrating tablets (ODTs) are a drug dosage form which are intended to rapidly disperse in the oral cavity. This dosage form is different from chewable or buccal tablets because it eliminates the need for prolonged presence in the mouth. The quick disintegration process addresses the needs of certain groups of patients who find it difficult to swallow intact tablets. Therefore, ODTs were designed to be used in pediatric, elderly, and non-compliance patients. However, ODTs have gained more attention as preferred solid drug dosage forms due to their increased convenience and compliance compared to conventional tablets and capsules [1]. These key trends appear to have fueled the market, which was estimated to have reached 10% of the compound annual growth rate in 2018 [2]. Drug manufacturers are turning to ODTs as a drug dosage form of choice for BCS (Biopharmaceuticals Classification System) class I and III compounds, where the challenge of delivering a drug does not lie in poor solubility. Highly soluble APIs (active pharmaceutical ingredients) along with rapidly disintegrating tablets provide faster dissolution, leading to the increased bioavailability of drugs and the quick onset of action. Recently, the industry not only successfully applied many ODTs formulations to introduce new delivery format to patients, formats such as Benadryl^®^ (Diphenhydramine), Lamictal^®^ ODT (lamotrigine), Zofran^®^ (Ondansetron), or Olanex Instab^®^ (Olanzapine) [2], but also introduced ODTs as an intended drug dosage formulation, e.g., Nurtec^®^ ODT (rimegepant) or Evekeo^®^ (amphetamine sulfate) ODT. The choice of excipients with rapidly disintegrating tablets is crucial for palatability; the optimization of the composition of ODTs is directed to obtain a clean mouth feeling or creaminess. The disintegration time can potentially be influenced by other quality attributes, such as the tensile strength (hardness) and porosity of tablets. In general, an increase in the hardness of a tablet lowers its porosity, leading to a slower disintegration time. Conversely, a lower than desired tensile strength can lead to chipping and breakage defects. Such defects impair the production and packaging processes and may affect the safety and effectiveness of the product, leading to therapeutic failure due to variances in its formulation. Optimizing the tensile strength of a tablet is dependent on several factors such as the choice of excipient, the physiochemical properties of powder material, the compaction force and speed, the moisture content, and the tablet dimension [3]. A formulation can have various components, such as disintegrants, lubricants, solubilizers, binders, and fillers. The choice of components is based on the dosage form requirements and the manufacturing process. Each component and its quantity have an impact on the overall critical quality attributes of a dosage form [4]. Understanding the directionality and magnitude of an effect that a formulation component might have on one or more CQAs is imperative for the optimization of the required characteristics of a dosage form, such as, e.g., disintegration time.

There are numerous choices of manufacturing processes to produce ODTs: lyophilization, molding, the cotton candy process, freeze- or spray-drying, mass extrusion, compaction, and other patented technologies [5]. These technologies require expensive equipment and processes and might have additional requirements for packaging compared to the standard, widely used direct compression manufacturing process, which involves fewer unit operations and widely accessible technology. However, ODTs produced by direct compression are exposed to high compression forces, leading to higher tensile strength, and as a result may exhibit high disintegration times. Therefore, when it comes to this dosage form, the challenge is to achieve a consistent structure that enables rapid disintegration without affecting the hardness of the tablets [6].

Developing a model for ODTs involves the interplay of formulation components, powder characteristics, and the manufacturing process to achieve desired product quality attributes. It has been a topic of modeling efforts in numerous previous studies. Such an approach aims to reduce the number of experiments, increase the understanding of diverse inputs, and move towards a quality by design (QbD) approach as defined by ICH Q8. Therefore, achieving the critical quality attribute of a short disintegration time (<180 s) [7] can be challenging given the complexity of the interplay between APIs, excipients, and tablet manufacturing process parameters.

Advances in ML have accelerated development in various fields including drug discovery and development. ML models are widely used to aide complex decision-making frameworks, with applications in early target biology (to search for viable and accurate disease targets), medicine design (to synthesize novel drug candidates) [8], and chemistry and manufacturing controls (to optimize formulations and critical process parameters to achieve critical quality attributes of a given dosage form) [9].

Predicting quality attributes for solid dosage forms has been a topic of consideration in many research endeavors, such as, for example, the prediction of the granule particle size distribution and the tensile strength of a tablet using ridge regression and random forests, respectively [10], the prediction of the capping of tablets using multivariate modeling tools [11], and prediction of tablet defects using convolutional neural networks [12]. ML modeling methods have also been applied to the formulation development of ODTs where multivariate tools and artificial intelligence approaches have been observed to perform well in the context of understanding the underlying relationships between the critical quality attributes and process parameters for ODTs [13]. While the above examples show that ML models can be trained to predict CQAs and prediction rules can be extracted, generalizability still remains a challenge. On the one hand, datasets have to be diverse in order to develop a model for various APIs and excipient combinations. On the other hand, the numerical encoding of chemical compounds has to be universal in the context of further application and knowledge discovery. In certain cases, the introduction of various compounds may lead to a decrease in predictive efficacy as a trade-off for data extrapolation.

A pre-formulation mathematical system called the SeDeM expert system is also widely used. It is reported to be suitable for individual and combinations of powders. The SeDeM system outputs a threshold value indicating whether a formulation can be manufactured by direct compression and if the resulting dosage form would be orally dispersible [14]. To utilize the expert system, a starter set of experimental readouts needs to be provided, based on which it interpolates the factor space of a given formulation and can provide suggestions for excipient ratios in the formulation. The system relies on experimental readouts, although it significantly reduces the overall number of experiments [15]. Currently, the SeDeM method is focused on the recommended formulation, but it cannot quantitatively predict the disintegration time of ODTs formulations. To address the challenge of pharmaceutical research, we need to establish a prediction method to help experts evaluate the performance of ODT formulations.

Han et al. [16] applied a neural network to predict disintegration times. Their models were trained on a dataset containing 145 formulation records covering 23 APIs, collected from the literature, and exhibited a prediction accuracy of 80% of the testing dataset [16]. This research endeavor reproduced the data collection from the literature and enhanced the dataset by auditing the source publications to remove errors, if any, and included molecular descriptors to describe APIs.

Empirical modeling methods have benefitted from advances in computational power and programming frameworks. It is common to train a variety of models on very large data sets without setting an *a priori* assumption of their structures and settings. Such models learn from labels along with examples presented to them as inputs. Modeling methods can be based on classification and regression trees, neural networks and deep learning, genetic programming and symbolic regression, and a combination of individual methods in an ensemble of models. Whilst complex ML models offer flexibility, as they can be trained for any task, and performance, as they can be accurate, precise, and fast, there is a trade-off for transparency against performance due to the black-box nature of ML models. This trade-off has not, on average, been a hindrance in most widespread applications of ML, including non-GxP tasks in the pharmaceutical domain; however, modeling for the purpose of optimizing a formulation and process parameters is required, by regulatory authorities, to be transparent and reproducible. In recent years, the ML community has endeavored to bring consistency, fairness, and transparency to modeling efforts through the use of best practices and explainable AI modeling (XAI) initiatives. A machine learning model is expected to be:

Trustworthy—the validity of the prediction can be assessed;Explainable—internal mechanisms to make prediction are clear;Usable—is effective, efficient and scalable;Transparent—understand aspects of the data that could influence predictions [17].

This effort aims to develop a variety of models to predict the disintegration times of ODTs using semi-automatic ML engines according to ML best practices and XAI techniques [18]. It is hoped that the results from model training, validation, and explainability will contribute towards domain understanding in the context of formulation design and the optimization of process parameters for manufacturing tasks.

## 2. Materials and Methods

### 2.1. Database–Data Scrapping

An existing literature-based data model [16] was selected for enhancement and curation. Data were filtered to include only complete and traceable data records with an emphasis on ODT quality attributes such as tablet hardness, thickness, and dimension of tablet press die. Records with omissions in quality attributes were not selected. To expand the database, a literature survey of Scopus^®^ (Elsevier, The Netherlands, https://www.scopus.com, accessed on 1 August 2021) was performed. Keywords such as “oral disintegrating tablets”, “orodispersible tablets”, and “disintegration time” were used. The resulting articles were reviewed to meet the following criteria: ODTs should be manufactured using the direct compression method; the amount of all excipients in the formulation should be present; tablet quality attributes (hardness, thickness) and parameters such as die dimension should be present; and the compendial disintegration test should be applied (Ph. Eur. or USP). In total, 29 articles, as shown in Table 1, fulfilled the data model requirements. The articles were the source of 256 unique data records (formulations), out of which 52 records were also present in the former database by Han et al. [16]. Fifty-two redundant records were extracted from eight overlapping articles. Overall, a new database consisting of 256 formulations covering 26 APIs was constructed and applied in modeling using ML methods. Each of the articles was the source of at least one data record (please see Table 1 ‘No of formulations’ column).

### 2.2. Data Enhancement, Preprocessing, and Exploratory Data Analysis (EDA)

According to the European Pharmacopoeia 10th ed., orodispersible tablets disintegrate within 3 min; therefore, all database records where the values of the ‘Disintegration time [s]’ variable exceeded 180 s were excluded from further analysis. Correlation analysis was conducted to assess the relationship between the dependent variable (disintegration time) and the independent variables (chemical descriptors, process parameters, composition, etc.). The inputs were enriched by calculating physical descriptors of the tablets, such as the lateral tablet surface area [mm^2^], flat tablet surface area [mm^2^], total tablet surface area [mm^2^], volume (tablet) [mm^3^], and surface area to volume ratio [mm]. Finally, APIs’ two-dimensional (2D) molecular descriptors were added using the mordred-descriptor v.1.2.1a1 Python package [48]. These molecular descriptors attempt to represent the detailed chemical structure of APIs. The pre-tableting parameters, used as inputs for models, were the punch die of tablet press [mm] and tablet mass [mg]. The post-tableting attributes, used as outputs to optimize the models, were thickness [mm] and hardness [N]. Tablet hardness was used as a surrogate of compression force because compression values were not available in all the selected publications. Moreover, to differentiate between different physicochemical properties of fumed silica (Aerosil) and other grades of colloidal silica, two separate variables were introduced, namely ‘Aerosil’ and ‘Colloidal silica’ [49].

### 2.3. State-of-the-Art ML Workflow 

The machine learning (ML) model development process had been divided into three general stages, as illustrated in Figure 1, namely: data pre-processing, modeling, and model interpretation. The stages follow well known best practices in ML where data augmentation and pre-processing are conducted by domain experts with the aim of representing the input material and process in the best possible manner [50], for example, the enhancement of a database by calculating the molecular descriptors of APIs and the physical characteristics of tablets. ML modeling tasks are designed to cover numerous types of modeling methods, including classification and regression trees and neural networks, whereby different models are trained using prepared database as an input, and an error metric is used as the selection criteria for the best models. Keeping track of all avenues of feature and model exploration can be challenging, as such tasks are computationally expansive; therefore, AutoML platforms are used. AutoML modeling methods are also designed to create model ensembles where different kinds of trained models are used collectively for prediction tasks [51]. In this endeavor, the H2O AutoML platform [52] was employed, which was set up to perform feature selection (based on a predefined threshold) and develop a final production model in the K-fold cross validation scheme. For the latter, K was set to 10, in order to increase granularity and reduce bias. Each fold consisted of a unique training–testing pair, which had randomly selected 218 or 219 records for training and 24 or 25 records for testing purposes. For specific training–testing pairs, please refer to the datasets at https://github.com/jszlek/ODT_database (accessed on 10 August 2021). The code used to reproduce the workflow and app can be accessed at https://github.com/jszlek/h2o_AutoML_Python (accessed on 10 August 2021) and https://github.com/jszlek/ODT_dash (accessed on 10 August 2021), respectively. An interactive tool has been published online on the Heroku server (https://odt-dash.herokuapp.com/) (accessed on 1 February 2022).

### 2.4. Model Training and Assessment

The training of ML models is conducted by learning a function that best represents the examples presented to it as inputs. The goal is to find a function that generalizes to a given task and can accurately predict unseen cases [53]. ML models are known to overfit, which is when a model learns to recognize the examples presented to it rather than generalize. To avoid overfitting and to ensure a robust model, feature selection is performed, whereby input features are selected based on their likelihood to contribute towards a generalized model [54]. Feature selection also offers insights into how a model makes predictions, and thus helps to demystify the black-box nature of ML models [55]. Furthermore, a portion of the data was retained to test the model in the k-fold scheme. Using a k-fold scheme ensures that the model has been trained and tested on all data in a fair manner over different iterations [56]. In this study, models were trained and tested in five-fold cross-validation splits for feature selection using a Python script. A randomly initialized seed was set to initiate values for the ML model hyperparameter space search. The training and testing step was repeated 25 times to ensure complete coverage of the input database to obtain the best model. Upon selection of the final input feature vector, a further 10-fold cross-validation scheme was used to train the final model. The accuracy and performance of the model were assessed using the root mean squared error (RMSE, Equation (1)), the normalized root mean squared error (NRMSE, Equation (2)), and the coefficient of determination (R^2^, Equation (3)). In order to assess the robustness of the final model, a multi-start technique was applied [57]. During feature selection and final model development, all available algorithms in the H2O AutoML platform were used, namely the distributed random forest (DRF), extremely randomized trees (XRT), generalized linear model (GLM), extreme gradient boosting machine (XGBoost), gradient boosting machine (GBM), deep learning (fully connected multilayer artificial neural network), and stacked ensemble models.
(1)$$\mathrm{RMSE}=\sqrt{\frac{\sum_{i=1}^{n}{\left(pred_{i}-obs_{i}\right)}^{2}}{n}}\;$$
(2)$$\mathrm{NRMSE}=\frac{RMSE}{obs_{max}-obs_{min}}\cdot 100\%\;$$
(3)$$R^{2}=1-\frac{SS_{res}}{SS_{tot}}=1-\frac{\sum_{i=1}^{n}{\left(pred_{i}-obs\right)}^{2}}{\sum_{i=1}^{n}{\left(obs_{i}-obs\right)}^{2}}$$
where *obs_i_*, *pred_i_* are the observed and predicted values, *i* is the data record number, and *n* is the total number of records, *obs_max_* is the maximal observed value, *obs_min_* is the minimal observed value, R^2^ is the coefficient of determination, *SS_res_* is the sum of squares of the residual errors, *SS_tot_* is the total sum of the errors, and *obs* is the arithmetical mean of observed values.

#### 2.4.1. Extremely Randomized Trees

The extremely randomized trees (XRT) method is derived from random forests and is based on the concept of decision tree ensembles [58]. Decision trees are a recursive partitioning algorithm which work by dividing training inputs into hierarchical groups to learn class labels. Such models are rarely used alone, since they are prone to overfitting. Random forests are bootstrapped ensembles of decision trees with the goal of reducing individual tree variance, whereby predictions from all trees in a random forest are averaged to give the final prediction [59]. XRT are a variant of random forests where the feature set is completely randomized to create hierarchies, but the entire database is used as a training set, as opposed to the creation of bootstrapped ensembles.

#### 2.4.2. Gradient Boosting

The gradient boosting (GB) method is also based on decision trees where each tree is built considering the error values in the previous one. Following such a scheme, the minimization of prediction errors can be achieved at each subsequent iteration using a learning algorithm such as a gradient descent [60,61]. GB methods are commonly used in classification and regression tasks.

#### 2.4.3. Feedforward Deep Neural Networks

A deep neural network is based on the concept of a multilayer perceptron (MLP). MLPs consist of single units known as neurons which contain an input layer, a transfer function, and an output layer. Multiple neurons can be connected in sequential layers to form a neural network. The task of training a neural network is to backpropagate errors from the output layer through the hidden layers using the gradient descent algorithm to optimize the weights of individual neurons working in parallel within a given layer [62]. Neural networks are known for high performance and have a wide range of applications [63]. 

### 2.5. Model Interpretation

ML models are mostly black-box in nature, but attempts can be made to explain their ways of generating predictions. Our workflow (Figure 1) utilizes two post-hoc model agnostic methods for model interpretation: the SHapley Additive exPlanations (SHAP) method by Lundberg et al. [64], which helps develop reasoning behind individual predictions of model, and partial dependency plots (PDP) [65], which are used to represent global relationships between input and output variables. The implementation of both methods was made using an in-house-developed Python wrapper script [66].

The SHAP method is a widely used approach from cooperative game theory that has been shown to be useful in the pharmaceutical domain [67]. The original concept of Shapely values was developed to calculate the contribution of an individual player towards the team effort [68]. While reasoning behind a single prediction might not offer much, collective reasoning behind a large set of predictions can help to detect global trends. When SHAP is applied for every data instance, a matrix of Shapley values is obtained where each row represents the data instance and each column represents a feature. Such global trends can be useful in providing hints to model working and diagnosing potential problems. The mathematical formula is represented by the following equation (Equation (4)):(4)$$\phi_{j}\left(val\right)=\sum_{S\subseteq \left\{1,\ldots ,p\right\}\setminus \left\{j\right\}}\frac{\left|S\right|!\left(p-\left|S\right|-1\right)!}{p!}\left(val\left(S\cup \left\{j\right\}\right)-val\left(S\right)\right)$$
where *S* is a subset of the features used in the model, *x* is the vector of feature values of the instance to be explained, and *p* is the number of features. *val_x_*(*S*) is the prediction for feature values in set *S* that are marginalized over features not included in set *S*.

The Shapley value calculation method satisfies the axioms of efficiency, symmetry, dummy, and additivity, which together provide the explanation of a prediction reasonable foundation. The values for each feature are replaced by drawing random instances to ascertain the importance and contribution of the features. Thus, Shapely value calculation is computationally intensive due to an exponentially large number of possible coalitions of the feature values. In such a scenario, coalitions are sampled to limit the number of iterations, thereby decreasing the computation time; however, the variance of the Shapley value increases. Therefore, a k-means method was used to reduce the number of iterations for explaining each feature contribution. The number of k-mean was set to 12, which is the number of centroids in a cluster representing each feature data domain. A manageable yet informative SHAP matrix can be obtained after clustering each feature data domain, which can be visualized to interpret model predictions.

Partial dependence is a conceptual extension of feature selection. While feature selection methods inform which features are important to a prediction, partial dependency plots can be used as a visualization method to further establish relationships between dependent and independent variables.

Representing functions of higher-dimensional arguments can be challenging. Partial dependence of the approximation of the function *f*(*x*) is used on selected small subsets of the input variables, which shows the marginal effect of one or two variables on the predicted value.
(5)$$\hat{f_{S}}\left(x_{S}\right)=E_{X_{C}}\left[\hat{f}\left(x_{S},X_{C}\right)\right]=\int \hat{f}\left(x_{S},X_{C}\right)dP\left(X_{C}\right)$$
where *x_S_* are the features for which the effect of prediction is required, hence the partial dependence function should be plotted. *X_C_* are other features treated as random variables. The combined feature vector for *x_S_* and *X_C_* constitutes the total feature space *x*. Partial dependence works by marginalizing the machine learning model output over the distribution of the features in set *C*, so that the function highlights the relationship between the features in set *S* and the predicted outcome. By marginalizing over the other features, we obtain a function that depends only on features in *S* and includes interactions with other features in set *C*. The partial dependence plot is a global method which considers all instances and gives insights into the global relationship of a feature with the predicted outcome.

## 3. Results

### 3.1. Database

Exploratory data analysis indicated that only one pair of features was correlated, namely: tablet mass [mg] with punch die of tablet press [mm] (Figure 2). This finding was expected, as a 0.5 ratio between tablet thickness and diameter is required to meet the basic hardness requirement for the tablet. Surprisingly, the absence of an expected linear relationship between disintegration time [s] and hardness [N] of tablets is observed. This may indicate that the hardness of most of formulations ensured their mechanical integrity. In general, there is no recommended range of hardness for ODTs; however, it is recommended for tablets produced by direct compression to maintain their hardness between 18 and 40 N for later packaging and handling. This observation is consistent with the observed distribution of hardness values in the database. Tablet mass, thickness, and punch die of tablet press also fall within the standard guidelines for ODTs [69]. Some tablets are observed to exhibit a larger size than usual (6–7 mm thickness, 16–20 mm diameter and 800–1200 mg tablet mass). Nevertheless, the disintegration time for the larger tablets was between 30–60 s, which is well within the recommended range for ODTs.

The preprocessed database consisted of 243 directly compressed ODTs formulations (unique data records), including 26 unique APIs. Each formulation was characterized by 633 molecular descriptors of APIs’, 28 variables encoding composition (excipients were encoded topologically), 9 variables encoding manufacture parameters (tablet mass [mg], thickness [mm], hardness [N], etc.), and the disintegration time [s]. The molecular weight of represented APIs ranged from 179 Da to 584 Da, with a median of 371 Da, and calculated logP (XLogP) values ranged from 1.14 to 10.61, with a median value of 3.54. The descriptive statistics before the inclusion of molecular descriptors are presented in Table 2. Violin plots with box plots (Figure 3), and descriptive statistics (Table 2) indicated that the variables were not normally distributed, the composition of formulations was especially positively skewed (right-skewed distribution), which was expected when topological encoding was applied. However, the performed split of the database according to 10-fold cross-validation was balanced to maintain the same distribution of input variables across the splits.

The preprocessed database and training/testing sets are available at https://github.com/jszlek/ODT_database (accessed on 1 March 2022).

### 3.2. Feature Selection and Final Model Development

The feature selection and final model development were performed in an automated procedure. The AutoML search parameters were set as presented in Table 3. The table represents the robustness of the models developed in a multistart procedure. The values of RMSE, NRMSE, and R^2^ are the average of 30 repetitions. The input variables were grouped into three main categories, namely composition, API’s molecular descriptor, and manufacturing parameter. Features from all categories were deselected below the variable importance threshold, except for features in the composition category. As a result, 39 features were included in the final input vector. Table 4 shows the selected features, categories, and relative scales of variable importance. The amount of disintegrants (croscarmellose sodium, crospovidone, sodium starch glycolate), manufacturing parameter (hardness), amount of solubilizer (Eudragit EPO) and lubricants (MgSt, Aerosil) are highly ranked, followed by molecular descriptors representing properties of an API (GATS7i, GATS7p, GGI7, etc.). Low ranking features include the amount of filler and binder in the formulation, e.g., lactose and sodium bicarbonate.

Based on the relative importance of the variables representing the amount of disintegrants (croscarmellose sodium, crospovidone, sodium starch glycolate) presented in Table 4, it is evident that the amounts of disintegrants will have the greatest impact on the predicted values. According to feature importance, the less important components of the tablets are the amounts of lubricants and some solubilizers (Eudragit EPO). The amounts of solubilizers are ranked lower in feature importance, which may have been caused by the positive skew of the distribution of the variables. 

The highest impact from the group of API’s molecular descriptors had GATS7i, GGI7, MATS4p, MIC2, XLogP, and GATS7p. GATS7i and GATS7p belong to the Geary autocorrelation with lag 7 descriptors, weighted by ionization potential or polarizability. These results suggest that the electric properties of an API may affect the disintegration time. Moreover, the GGI7 descriptor is the topological charge index of order 7, and it may also be considered useful for describing the charge location inside the molecule. Another descriptor related to electrical properties is MATS4p, the Moran autocorrelation of lag 4 weighted by polarizability. The last two descriptors, XLogP (theoretical n-octanol–water partition coefficient) and MIC2 (modified information content index, neighborhood symmetry of 2-order) are related to the lipophilic–hydrophilic balance and geometry of the molecule, respectively. The modified information content (MIC2) index is weighted by mass and carries information on the topology of a molecule and the multiplicity of bonds around individual atoms. The increase in MIC2 is observed when additional bonds are introduced (e.g., branched vs. linear carbon chain isomers). The discussion on the variables nT12Ring and nF8HeteroRing has been deliberately omitted due to the poor representation of these descriptors in the database.

Interestingly, in the input vector for final model development, tablet mass and manually calculated surface area parameters were not included. As the exploratory data analysis (Figure 2) indicated, the tablet mass was linearly correlated with the punch die of tablet press, therefore it could be discarded without loss of information and degradation of the overall performance of the model.

The best final model yielded a 10-fold cross-validation RMSE of 10.92 (NRMSE = 8.1%) and an R^2^ of 0.84. It consisted of 39 inputs, which were selected according to the variable importance. The deep neural network model had four layers: a 39 neurons input layer, two 100 neurons hidden layers, and a one neuron output layer. The hidden layers had a rectifier with a dropout activation function. The neural network was trained for 3341 epochs.

### 3.3. Model Explanation

The SHAP summary plot represents how the features affect the output of the model’s prediction. The color bar depicts the actual features values, where on the x-axis, an impact (positive or negative) on a prediction can be observed (Figure 4). By means of a SHAP summary plot, general effects and assumptions could be drawn.

Higher disintegration times are predicted where higher amounts of disintegrants (crospovidone, croscarmellose sodium, sodium starch glycolate) occur. A similar effect can be observed for fillers (MCC, Mannitol). However, in the case of lactose, the opposite relationship is observed, where a higher amount of lactose leads to lower disintegration times. For lubricants (Aerosil, MgSt, and SSF), two separate effects can be observed: high amounts of Aerosil and SSF lead to higher disintegration times, probably due to the hydrophilic nature of those excipients. However, MgSt (more lipophilic) reduces the disintegration times when it is present in high quantities because of the occlusive effect. Solubilizers (Eudragit EPO, SLS), in small amounts, do not slow down disintegration. In the case of camphor, its higher content contributed to faster disintegration, likely due to the function of this excipient. Camphor is used as a porophore, and its amount is directly related to the porosity of a tablet.

A closer inspection of the stacked SHAP plots was carried out to identify the global effects of three groups of features, namely formulation composition, manufacturing parameters, and API’s molecular descriptors.

The results presented in Figure 5 reflect the general view on the influence of individual features on the disintegration time of ODTs. Figure 5A–C depict the effects exerted by super-disintegrants on the average model’s prediction (37.5 s). The trends shown in SHAP plots suggest that higher amounts of super-disintegrants lead to faster disintegration times, with crospovidone being the most universal excipient in lowering the disintegration time (Figure 5A). Even small amounts of crospovidone (more than 3%) can accelerate tablet disintegration by up to 15 s. In case of other super-disintegrants, CC-Na and SSG, the plots (Figure 5B,C) are jagged, and this can indicate the inconclusive average effects of those excipients on disintegration time, which might be influenced by other constituents of a tablet. Furthermore, concerning the effects of lubricants on the average model’s prediction, it could be observed that the addition of Aerosil (more than 0.5%) accelerates the disintegration time, probably by increasing the overall hydrophilicity. However, when MgSt is used in amounts greater than 1%, the overall hydrophilicity drops due to the film forming effect, and as a result the disintegration time increases. The effects on the average prediction for Eudragit EPO indicate that very small (<1%) and very high (>20%) amounts of solubilizer decrease the disintegration time, while moderate concentrations (1–20%) can favor slower disintegration. This phenomenon is also observed for PVP (data not shown). These observations may indicate that if low and high concentrations of polymer solubilizers are used, the mechanism of disintegration occurs through wicking; meanwhile, in case of moderate amounts of polymers, the swelling mechanism prevails.

Figure 6 depicts effects of manufacturing parameters on disintegration time, as predicted by our best model. Not surprisingly, these predictions reflect the FDA guidelines for ODTs. The following tablet attributes adversely affect the disintegration time of ODTs: punch die of tablet press larger than 12 mm (Figure 6A), thickness of more than 4.5 mm (Figure 6B), hardness greater than 100 N (Figure 6C), and amount of API greater than 15% (Figure 6D). When accumulated, these effects, although not clearly marked, may adversely affect the formulation.

Figure 7 shows the influence of individual API’s molecular descriptors on the disintegration time. Descriptors representing the distribution of electric charges (GATS7i, GATS7p, GGI7, MATS4p) can be divided into two groups. The first group is distinguished by the fact that when the value of molecular descriptors decreases, the disintegration also decreases (GATS7i and MATS4p, Figure 7A,D). However, the effect of this transition is not very profound (the decrease is estimated to be 2–3 s from the average prediction). On the other hand, the effect on disintegration time is the opposite of this for the second group (GATS7p and GGI7, Figure 7B,C); with an increase in the descriptor value, the disintegration time tends to decrease. The same could be noticed for the other two descriptors, MIC2 and XLogP (Figure 7E,F). The increase in the MIC2 descriptor value, related to the topological structure of a compound, is responsible for the decrease in the average disintegration time. On the contrary, increased lipophilicity (XLogP) slows down disintegration.

Three-dimensional plots of partial dependence were used in order to investigate the influence of APIs’ XLogP vs. type and amount of super-disintegrant and lubricant. Figure 8A–C depicts how the lipophilicity of APIs’ impacts the disintegration time when different types of super-disintegrants are used. It can be noted that disintegration is most accelerated in the case of crospovidone (Figure 8A), which is weakly affected by the lipophilicity of APIs. A gradual decrease in disintegration time with an increase in super-disintegrant content is observed. Moreover, for highly lipophilic compounds (XLogP > 5), higher amounts of crospovidone (>10%) are required to obtain disintegration times of less than 35 s. The remaining super-disintegrants tend to have less disintegration activity. Figure 8B,C shows the unfavorable combination of super-disintegrant and XLogP, where the disintegration time is maximized. Based on Figure 8A–C, the listed compounds can be ranked in order of best performance against a broad spectrum of API lipophilicity over the time of disintegration: crospovidone > SSG > CC-Na. 

Similar effects can be noticed in Figure 8D–F for lubricants. The most effective lubricant in terms of reducing the disintegration time in a mixture of compounds with different lipophilicity is Aerosil. However, compared to other lubricants, relatively high amounts (>3%) of these excipients should be added to achieve the desired effect. On the contrary, this amount of Aerosil is sufficient regardless of the APIs’ lipophilicity. Magnesium stearate (MgSt, Figure 8E) can be characterized by contrasting properties. An increase in the amount of this excipient gradually increases the disintegration time. According to Figure 8F, sodium stearyl fumarate has a transient characteristic. It decreases the disintegration time when introduced to mixtures with hydrophilic APIs (XLogP~2–4). However, the effect is weaker for highly lipophilic compounds.

### 3.4. Software

The model and software for the prediction of ODT disintegration times were published on the GitHub server at https://github.com/jszlek/ODT_dash (accessed on 10 August 2021). The software runs on any machine with Python 3.6+ and additional libraries installed. Moreover, a fully functional online application was deployed on the Heroku server (https://odt-dash.herokuapp.com) (accessed on 1 February 2022).

## 4. Discussion

The disintegration process is an essential quality attribute that ensures the preparation of ODTs. Not only do the palatability of the tablet and the compliance of patients rely on it, but so too does the dissolution and bioavailability of the API. The disintegration of a tablet is the result of weakening interparticulate forces. In general, a disintegration process is caused by several factors, but it is primarily fueled by the penetration of water into a tablet [70], which is influenced by a combination of porosity and capillary action. When liquid imbibes (wicking) into powder compacts, three mechanisms are proposed to explain the weakening of particle-particle bonds: swelling, strain recovery (repulsive forces), and the dissolution of the soluble components of powder compacts. The presence of super-disintegrants usually facilitates the degradation of structural integrity. However, due to physicochemical differences in tablet components, the direct effect on the disintegration time in complex mixtures is uncertain. Shah and Augsburger [71] confirmed that physical differences in terms of super-disintegrants affect the disintegration time depending on the type of filler, even though the general action of a super-disintegrant is to accelerate disintegration. However, as shown in Figure 8, other components of the formulation can impede disintegration. In this case, the disintegration time modifying parameter is the lipophilicity of APIs (XLogP). Similar effects to the ones presented in this paper were also observed by Shah and Augsburger [71]. The most effective disintegrant, regardless of the XLogP values of APIs, was crospovidone. This phenomenon can be explained when the mechanism of tablet disintegration is analyzed. Crospovidone enhances wicking (absorption of water), but it is a non-swellable polymer, and therefore it is postulated that ODTs disintegrate by the means of a strain recovery (repulsive force) mechanism, regardless of the nature of an API (the polymer fulfils its role both for lipophilic and hydrophilic compounds). On the other hand, CC-Na and SSG do not influence wicking, but these polymers are highly swellable and act from the surface to the core of a tablet [71]. Therefore, an API or excipients of high lipophilicity can impair the inflow of water and the overall disintegration time. Lubricants are another example of excipients, which hydrophilic–lipophilic properties affect the disintegration process. ODTs manufactured by direct compression usually contain a hydrophobic component (MgSt, Talc), which is used to increase powder flowability and reduce the ejection force. However, the addition of large amounts of MgSt may alter the penetration of liquid into a tablet [72]. In fact, hydrophobic lubricants with an amount of more than 1% will lead to an increase in the disintegration time, through their film forming activity on other excipients [73].

The disintegration time can be affected by the presence of solubilizers. The mechanism of such actions can be directly connected to their function, which is to improve solubilization, act as a wetting agent for high loads of API, and increase both dissolution and bioavailability. Hence, the mechanism following the use of solubilizer in ODTs would be an increase in wettability and probably an increase in wicking through the higher and faster dissolution of an API, followed by the formation of pores. However, a moderate amount of wetting agent may increase the diameter of pores, so that the wicking capacity (capillary force) will be retarded, which will then slow down the process of disintegration [74].

Moreover, process parameters such as the hardness of a tablet, which is a surrogate of compression force, inflicts internal porosity and adhesion forces between particles. When the hardness is increased above 100 N, according to Figure 6, the pores are smaller and the adhesion forces are greater, and, as a result, the penetration of liquid is slower. On the other hand, a hardness lower than 100 N does not directly lead to faster disintegration. In other words, the effect is not related only to hardness. According to the strain recovery (repulsive force) theory, if the compaction force is too low, the pores are loose and the fluid is unable to penetrate into a tablet because the wicking is impaired due to the lower capillary forces [70,75,76]. Therefore, below a certain threshold (<100 N) of hardness, slower disintegration could be observed.

In general, an increase in the lipophilicity (increase of the XLogP value) descriptor value (increase lipophilicity) increases the disintegration time. This phenomenon is intuitively explained by lower water absorption and wettability due to impaired API solubility [77]. However, Fukami et al. [78], Yoshihashi et al. [79], and Iwao et al. [80] have proved that, in addition to lipophilicity, the physicochemical properties of a tablet’s components, such as the distribution of the charge in the molecule (polarity) and the free surface area, can influence the disintegration time. Interestingly, the results demonstrated in Figure 7 suggest that the increased polarity and topological properties (GATS7p, GGI7, MIC2) of an API can improve disintegration. This effect can be explained by the repulsive theory, where the force separating two faces with opposite properties (hydrophobic and hydrophilic) acts as the wicking proceeds [78]. 

## 5. Conclusions

The rapid development of artificial intelligence and machine learning tools, together with the increasing computational capacity of modern computers, creates a great opportunity for the pharmaceutical industry. These changes are also visible in the form of regulators such as the FDA, which developed guidelines and pilot programs targeting the application of AI/ML in healthcare. Within FDA initiatives, the development of model-based products is worth mentioning. Concepts such as model-informed drug development (MIDD) and machine learning have been noticed by the agency [81]. The areas in which predictive models can be used include the understanding of the production process and knowledge discovery.

A data-driven modeling paradigm, which the current AI/ML is based on, demands both high quality and large quantities of data. The former ensures precision (high predictability), whereas the latter accounts for the scope of the developed models. Given the highly automated manner of contemporary AI/ML implementations, the search for crucial variables and the handling of missing data via, for example, data imputation, has also become a domain of AI/ML. Having that said, we presume that when the dataset can be extended both in number of cases and features, the resulting models retain or improve their efficacy yet broaden their scope. The quantity of data could be also a factor in the context of improving the handling of incomplete features, when remaining cases would provide the means for data imputation. This is, of course, case-related, yet AI/ML works surprisingly well when it comes to filling the holes in the data when provided with a large number of cases to analyze.

ODTs manufactured by a direct compression process are complex systems. Many factors affect critical quality attributes, including the tablet disintegration time. In the present study, a new database focused on disintegration time was created from the literature, which was then utilized to develop the ML model using the H2O AutoML platform. An explainability analysis was carried out to understand the foundations underlying the final model’s predictions using Shapley values and partial dependency plots. Our findings on the effect that various formulation components exert on the disintegration time are corroborated by the existing literature and experts, showing that AutoML-based approaches are suitable for modeling complex pharmaceutical tasks. However, ML is constrained by the availability of data; thus, such models can be improved by the extension of well-structured and labeled datasets. 

The source data, scripts used in this work, together with the online version of the model, are freely available, as stated in the Appendix A section.

## Figures and Tables

**Figure 1 pharmaceutics-14-00859-f001:**
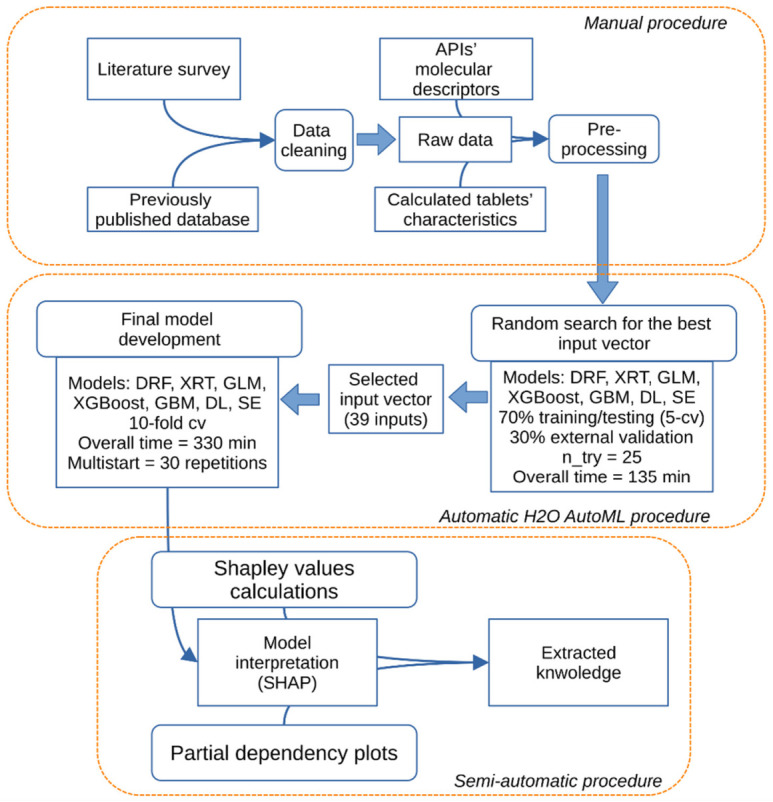
Schematic representation of the applied workflow. Models: distributed random forest (DRF), extremely randomized trees (XRT), generalized linear model (GLM), extreme gradient boosting machine (XGBoost), gradient boosting machine (GBM), deep learning (fully connected multilayer artificial neural network, DL), and stacked ensemble (SE); n_try, number of starting points for probing hyperparameter space; cv, cross-validation; API, active pharmaceutical ingredient.

**Figure 2 pharmaceutics-14-00859-f002:**
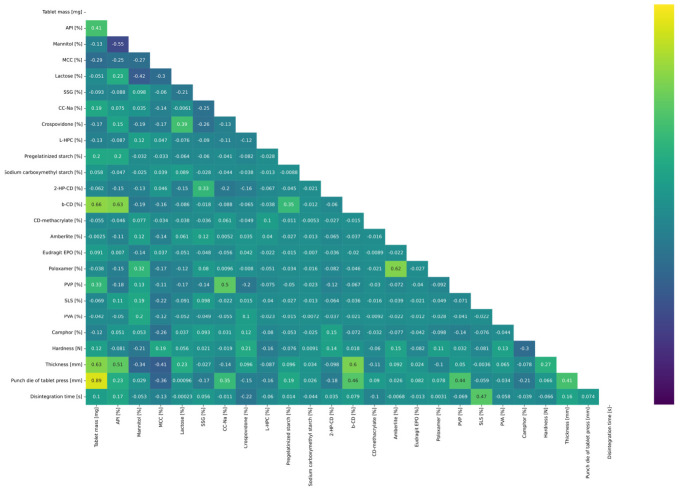
Correlation matrix of the database.

**Figure 3 pharmaceutics-14-00859-f003:**
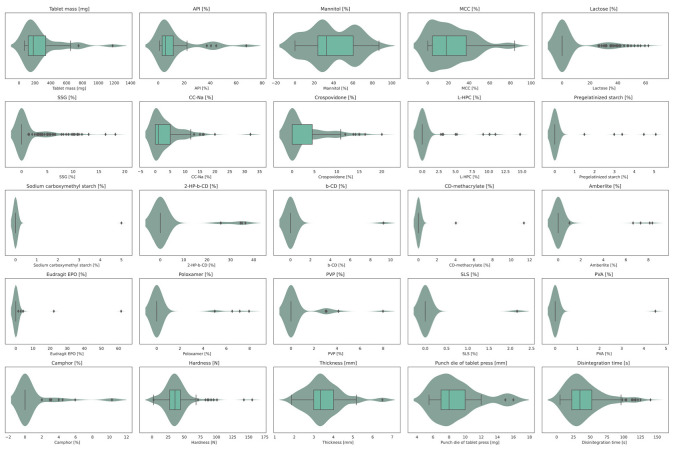
Box and violin plot of selected variables present in the database. Boxes represent interquartile range (IQR), namely: first quartile (Q1), median (horizontal line), third quartile (Q3), and the lower whisker = Q1–1.5*IQR; the higher whisker = Q3 + 1.5*IQR; curves represent distributions of numeric data using kernel density function.

**Figure 4 pharmaceutics-14-00859-f004:**
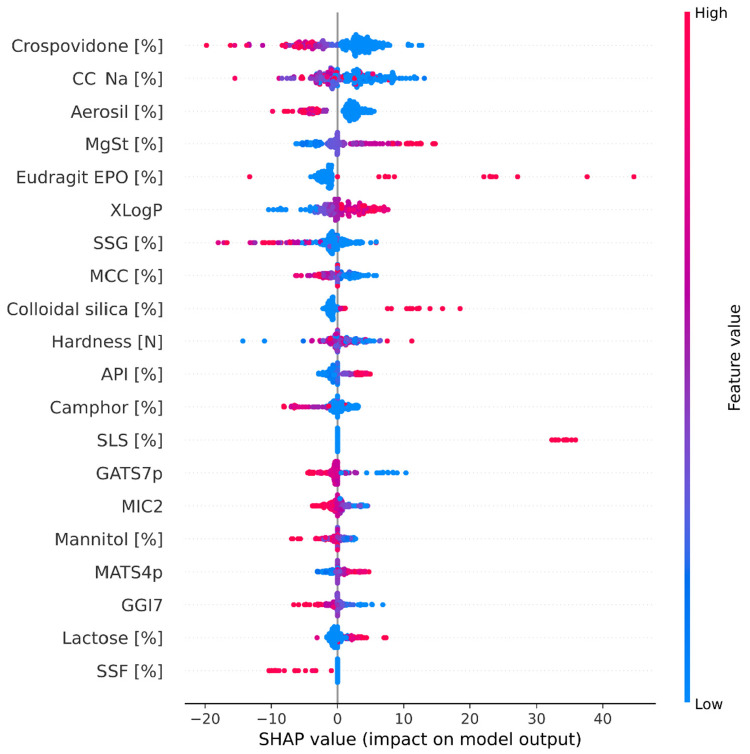
SHAP dependence plot of the top 20 features of the deep learning model. MCC, microcrystalline cellulose; CC-Na, croscarmellose sodium; SSG, sodium starch glycollate; MgSt, magnesium stearate; SSF, sodium stearyl fumarate; API, active pharmaceutical ingredient.

**Figure 5 pharmaceutics-14-00859-f005:**
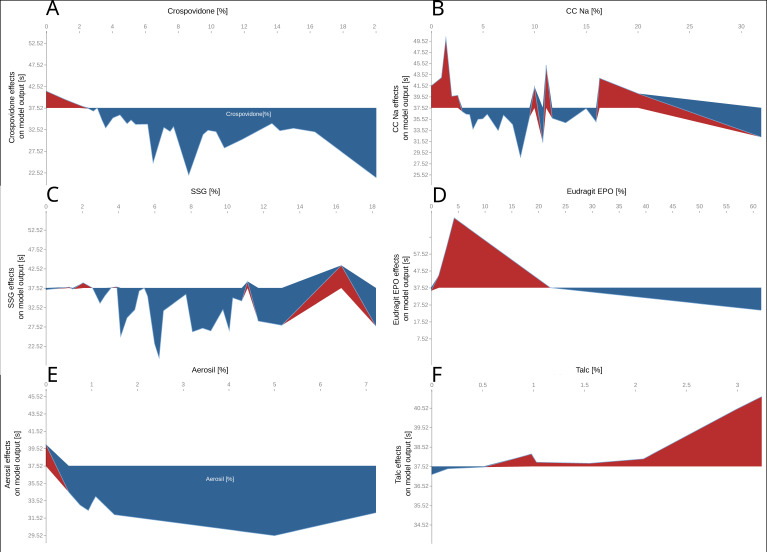
SHAP plots representing the effects of formulation composition on the disintegration time [s] for: Crosspovidone [%] (**A**), croscarmellose sodium (CC-Na) [%] (**B**), sodium starch glycolate (SSG) [%] (**C**), Eudragit EPO [%] (**D**), Aerosil [%] (**E**), Talc [%] (**F**).

**Figure 6 pharmaceutics-14-00859-f006:**
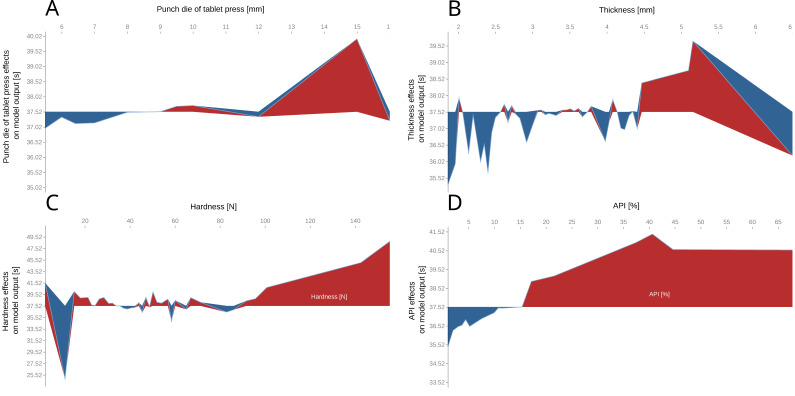
SHAP plots representing effects of various manufacturing parameters on the disintegration time: punch die of tablet press [mm] (**A**), thickness [mm] (**B**), hardness [N] (**C**), amount of API [%] (**D**).

**Figure 7 pharmaceutics-14-00859-f007:**
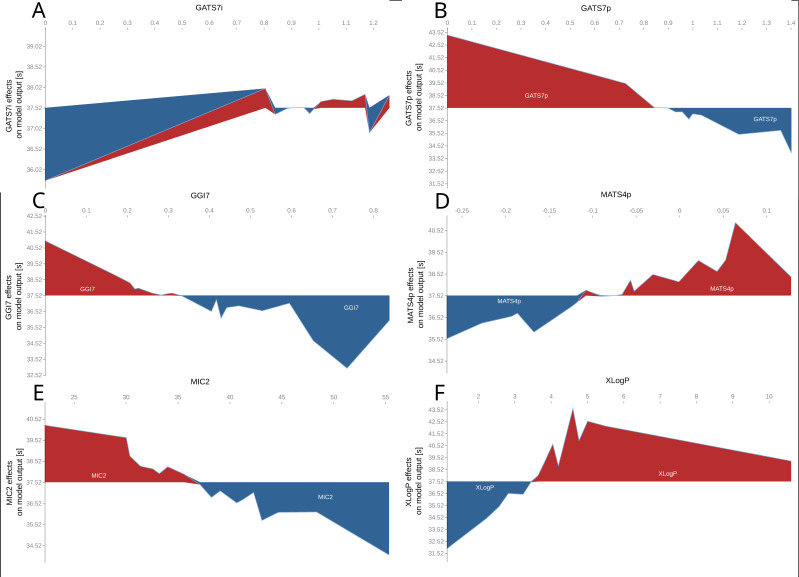
SHAP plots representing the effects of APIs molecular descriptors on disintegration time. GATS7i and GATS7p, the Geary autocorrelation with lag 7 descriptors, weighted by ionization potential (**A**) or polarizability (**B**); GGI7, the topological charge index of order 7 (**C**); MATS4p, the Moran autocorrelation of lag 4 weighted by polarizability (**D**); MIC2, a modified information content index, neighborhood symmetry of 2-order descriptor (**E**); XLogP, a theoretical n-octanol–water partition coefficient (**F**).

**Figure 8 pharmaceutics-14-00859-f008:**
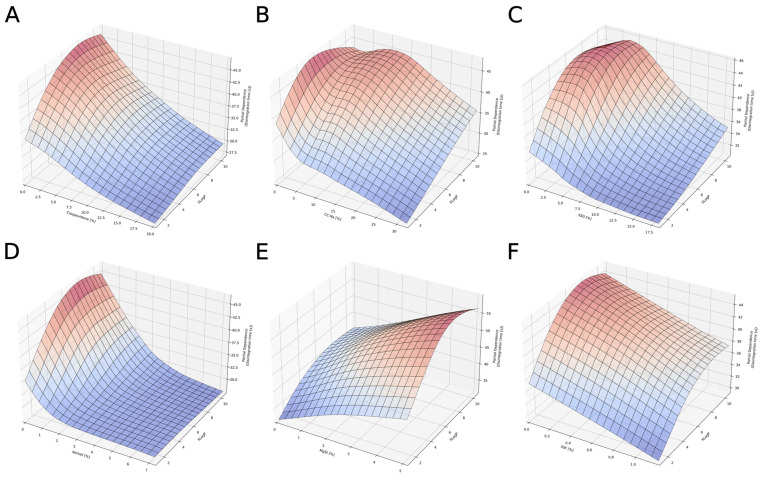
Partial dependency plots for XLogP vs. super-disintegrants: crospovidone (**A**), CC-Na (**B**), SSG (**C**), and lubricants: Aerosil (**D**), MgSt (**E**), SSF (**F**). XLogP, a theoretical n-octanol–water partition coefficient; SSG, sodium starch glycolate; MgSt, magnesium stearate; SSF, sodium stearyl fumarate; CC-Na, croscarmellose sodium.

**Table 1 pharmaceutics-14-00859-t001:** Source of data–publications [19,20,21,22,23,24,25,26,27,28,29,30,31,32,33,34,35,36,37,38,39,40,41,42,43,44,45,46,47].

API	Dose [mg]	Filler	Binder	Disintegrant	Lubricant	Solubilizer	No. of Formulations	Reference
Aceclofenac	100	Lactose, MCC	-	CC-Na	MgSt	-	9	[19]
Carbinoxamine maleate	4	Mannitol, MCC	-	L-HPC	MgSt	Amberlite	5	[20]
Carvedilol	12.5	Mannitol, MCC	-	SSG	MgSt, Talc	2-hydroxypropyl-β-cyclodextrin, Camphore-as a porophore	15	[21]
Dexamethasone	2	Mannitol, Lactose, MCC	-	Crospovidone	MgSt, Colloidal sillica	-	13	[22]
Dextromethorphan	15	Mannitol, Lactose, MCC	-	-	MgSt	Amberlite	2	[23] *
Donepezil	10	Mannitol	-	Crospovidone, CC-Na, SSG	Sodium stearyl fumarate	Poloxamer, Amberlite	6	[24] *
Drotaverine HCl	40	Mannitol	Calcium silicate, HPMC	Crospovidone, CC-Na	MgSt	PVP	20	[25]
Eletriptan	20	Mannitol, MCC		CC-Na, SSG, Crospovidone	MgSt, Talc	-	9	[26]
Eslicarbazepine	800	Mannitol, MCC	-	Crospovidone, SSG, Pregelatinized starch	MgSt, Talc	β-cyclodextrin	8	[27] *
Glipizide	10	Mannitol, MCC		CC-Na, SSG, Crospovidone, Pregelatinized starch	MgSt, Aerosil, Talc	-	9	[28]
Granisetron HCl	50	Mannitol, MCC	-	Crospovidone, CC-Na, SSG	MgSt, Aerosil	-	6	[29]
Granisetron HCl	2.4	Mannitol, MCC	-	CC-Na, SSG, Crospovidone	MgSt, Talc	Camphore–as porophore	12	[30]
Loratadine	10	Mannitol	-	Crospovidone, CC-Na	MgSt	PVA	6	[31]
Lornoxicam	4	Mannitol, MCC	-	CC-Na, L-HPC	MgSt, Aerosil	Cyclodextrin methacrylate	3	[32] *
Lornoxicam	8	Mannitol	-	Crospovidone, SSG, Pregelatinized starch	MgSt	-	4	[33]
Mefenamic acid	100	MCC	-	Crospovidone	MgSt, Aerosil	Eudragit EPO	2	[34] *
Meloxicam	7.5	Mannitol, Lactose, MCC	-	Crospovidone	MgSt	-	1	[35] *
Memantine HCl	5	Mannitol, MCC	-	CC-Na	MgSt, Colloidal silica	Eudragit EPO	15	[36]
Memantine HCl	10	Mannitol, MCC	-	CC-Na	MgSt, Aerosil	-	3	[37]
Montelukast sodium	5.2	Mannitol, MCC	Sodium bicarbonate	Crospovidone	MgSt	-	8	[38]
Mosapride citrate	5	Mannitol, Lactose, MCC	-	CC-Na, Sodium carboxymethyl starch, L-HPC, Crospovidone, Pregelatinized starch	MgSt	-	7	[39]
Olanzapine	10	Mannitol, MCC	-	SSG, CC-Na, Crospovidone	MgSt, Aerosil	2-hydroxypropyl-β-cyclodextrin	10	[40] *
Ondansetron	8	Mannitol, MCC	-	Crospovidone, CC-Na, SSG, L-HPC	SSF, Aerosil	-	20	[41] *
Propafenone HCl	150	Lactose	-	Crospovidone, CC-Na	MgSt	Camphore–as porophore	15	[42]
Propranolol HCl	40	Mannitol	-	Crospovidone, CC-Na, SSG	MgSt, Talc	SLS	9	[43]
Salbutamol suphate	4	Mannitol, MCC	-	CC-Na, SSG	MgSt, Talc	-	7	[44]
Simvastatin	5	Mannitol, MCC	-	CC-Na	MgSt	Poloxamer	9	[45]
Tadalafil	5	Mannitol, MCC	-	CC-Na	Talc	PVP	5	[46]
Ziprasidone HCl Monohydrate	22.63	Mannitol, MCC	-	CC-Na	MgSt	PVP	18	[47]

MCC, microcrystalline celulose; CC-Na, croscarmelose sodium; SSG, sodium starch glycollate; L-HPC, low-substituted hydroxypropylcellulose; HPMC, hydroxypropylcellulose; PVP, polyvinylpyrrolidone; PVA, polyvinyl alcohol; SLS, sodium lauryl sulfate; MgSt, magnesium stearate; SSF, sodium stearyl fumarate; *—formulations formerly present in database by Han et al. [16].

**Table 2 pharmaceutics-14-00859-t002:** Descriptive statistics of the database. MCC, microcrystalline cellulose; CC-Na, croscarmellose sodium; SSG, sodium starch glycollate; L-HPC, low-substituted hydroxypropyl-cellulose; PVP, polyvinylpyrrolidone; PVA, polyvinyl alcohol; SLS, sodium lauryl sulfate; MgSt, magnesium stearate; SSF, sodium stearyl fumarate; API, active pharmaceutical ingredient; 2-HP-beta-CD, 2-hydroxypropyl-beta-cyclodextrin; CD-methacrylate, beta-cyclodextrin-methacrylate.

Variable	Count	Mean	Std	Min	25%	50%	75%	Max
Tablet mass [mg]	243	274.10	252.29	67.13	116.4	180	336	1179.98
API [%]	243	12.81	16.25	1	3.02	5.56	11.59	67.8
Mannitol [%]	243	37.76	24.24	0	23.7	32.61	60.35	86.84
MCC [%]	243	22.62	20.37	0	4.57	18.12	37.32	84.1
Lactose [%]	243	7.19	15.45	0	0	0	0	62
SSG [%]	243	1.35	3.08	0	0	0	0	18.21
CC-Na [%]	243	3.43	4.99	0	0	1	5	31.95
Crospovidone [%]	243	2.55	4.28	0	0	0	4.5	20.03
L-HPC [%]	243	0.40	1.94	0	0	0	0	14.71
Pregelatinized starch [%]	243	0.07	0.53	0	0	0	0	5.08
Sodium carboxymethyl starch [%]	243	0.02	0.32	0	0	0	0	5
2-HP-beta-CD [%]	243	3.12	9.56	0	0	0	0	36.46
beta-CD [%]	243	0.31	1.66	0	0	0	0	9.31
CD-methacrylate [%]	243	0.06	0.77	0	0	0	0	11.39
Amberlite [%]	243	0.27	1.38	0	0	0	0	8.35
Eudragit-EPO [%]	243	0.46	4.22	0	0	0	0	61.54
Poloxamer [%]	243	0.37	1.46	0	0	0	0	7.95
PVP [%]	243	0.55	1.51	0	0	0	0	7.99
SLS [%]	243	0.08	0.41	0	0	0	0	2.16
PVA [%]	243	0.06	0.50	0	0	0	0	4.52
Camphor [%]	243	0.97	2.50	0	0	0	0	10.31
Hardness [N]	243	36.58	18.98	2.4	27.415	35.69	44.075	155.43
Thickness [mm]	243	3.50	0.93	1.86	2.995	3.34	4.01	6.5
Punch die of tablet press [mm]	243	8.86	2.86	5.5	7	8	10	16
Disintegration time [s]	243	41.13	27.35	4.98	22.5	34.66	52.34	140

**Table 3 pharmaceutics-14-00859-t003:** Hyperparameters and robustness of the H2O AutoML model development (multistart); mean values of RMSE, NRMSE, and R^2^ are provided for the developed models in a multistart procedure with 30 repetitions; standard deviation is in round brackets. DRF, distributed random forest; XRT, extremely randomized trees; GLM, generalized linear model; XGBoost, extreme gradient boosting machine; GBM, gradient boosting machine; DL, deep learning (fully connected multilayer artificial neural network); SE, stacked ensemble.

Repetition	Hyperparameter Search	RMSE [s]	NRMSE [%]	R^2^
30	Feature selection short loop time = 180 s Feature selection = 1 h No. of feature selection short loops = 25 Feature selection variable threshold = 0.1 Final model development (10-fold cv) short loop time = 120 s Final model development (10-fold cv) = 4 h No. of final model development (10-fold cv) short loops = 45 No. of cross validation folds = 10 All available models (DRF, XRT, GLM, XGBoost, GBM, DL, SE)	11.37 (0.42)	8.42 (0.31)	0.83 (0.01)

**Table 4 pharmaceutics-14-00859-t004:** Selected input vector for the best predictive model.

Variable	Variable Type	Scaled Variable Importance
CC-Na [%]	Composition, disintegrant	1.0000
Crospovidone [%]	Composition, disintegrant	0.8013
SSG [%]	Composition, disintegrant	0.7341
Hardness [N]	Manufacturing parameter	0.6564
Eudragit EPO [%]	Composition, solubilizer	0.5620
MgSt [%]	Composition, lubricant	0.5008
Aerosil [%]	Composition, lubricant	0.3991
GATS7i	API molecular descriptor	0.3441
MCC [%]	Composition, filler	0.3394
Colloidal silica [%]	Composition, lubricant	0.2336
Mannitol [%]	Composition, filler	0.2335
Pregelatinized starch [%]	Composition, disintegrant	0.2009
PVA [%]	Composition, solubilizer	0.1618
Thickness [mm]	Manufacturing parameter	0.1482
CD-methacrylate [%]	Composition, solubilizer	0.1253
GGI7	API molecular descriptor	0.1168
MATS4p	API molecular descriptor	0.1148
MIC2	API molecular descriptor	0.1133
API [%]	Composition	0.1109
Punch die of tablet press [mm]	Manufacturing parameter	0.1058
nT12Ring	API molecular descriptor	0.1053
XLogP	API molecular descriptor	0.1048
GATS7p	API molecular descriptor	0.1046
nF8HeteroRing	API molecular descriptor	0.1038
Amberlite [%]	Composition, solubilizer	0.0972
Sodium carboxymethyl starch [%]	Composition, disintegrant	0.0955
SLS [%]	Composition, solubilizer	0.0952
Camphor [%]	Composition, solubilizer (porophore)	0.0896
Calcium silicate [%]	Composition, binder	0.0868
Poloxamer [%]	Composition, solubilizer	0.0862
Sodium bicarbonate [%]	Composition, binder	0.0839
beta-CD [%]	Composition, solubilizer	0.0831
Talc [%]	Composition, lubricant	0.0830
2-HP-beta-CD [%]	Composition, solubilizer	0.0816
SSF [%]	Composition, lubricant	0.0751
HPMC [%]	Composition, binder	0.0675
Lactose [%]	Composition, filler	0.0591
L-HPC [%]	Composition, disintegrant	0.0542
PVP [%]	Composition, solubilizer	0.0525

## Data Availability

https://github.com/jszlek/ODT_database—Database and the training/testing pairs for 10-fold cross-validation.

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
