# Peer review of "Puzzle out Machine Learning Model-Explaining Disintegration Process in ODTs"

_pharmaceutics, 2022, doi:10.3390/pharmaceutics14040859_

Round 1
Reviewer 1 Report
Dear authors,
thank you for the manuscript.
I think it might be interesting for the Journal's readership.
However, I have a few concerns regarding the text. They are provided below.
1. Could you please provide some successful examples of marketed drugs having active pharmacological ingredients that use ODT as formulations?
2. Table 1 contains links on 29 APIs from literature. But how 256 API were obtained? If 145 unique records were retrieved from "the original database by Han et al. [16]" and about 29 were retrieved from literature, then how it can give 256 in total? How many unique samples were used for training? There are a lot of questions that need answer.
2. Table 3 contains a mix of hyperparameters for various machine learning methods. For which method RMSE is provided? Is it average RMSE? Please, specify.
3. Please explain physical meaning of selected features and their possible impact on the disintegration time of ODTs.
Minor:
- "It is common to train a variety of models on very large
data sets without setting a priori". The phrase looks unclear. - Figures need more readable text and axex caption.
Author Response
Dear Editor and Reviewers,
We were pleased to have an opportunity to revise our manuscript entitled “Puzzle out machine learning model - explaining disintegration process in ODTs” (1653469). In a revised manuscript, we have carefully considered reviewers’ comments and suggestions. As instructed, we have attempted to succinctly explain changes made in reaction to all comments. We reply to each comment in point-by-point fashion. We have color coded revised manuscript as text. The responses to the concerns raised by reviewers are below and are color coded as follows: a) Comments from reviewers are shown as text; b) Our responses are shown as text; c) Changes in the manuscript according to reviewers’ suggestions are highlighted.
The reviewers’ comments were very helpful overall, and we are appreciative of such constructive feedback on our original submission. After addressing the issues raised, we feel the quality of the paper is much improved. We hope that the manuscript in its present form is suited for publication.
Sincerely,
On behalf of all authors,
Jakub Szlęk
Reviewer #1
1. Could you please provide some successful examples of marketed drugs having active pharmacological ingredients that use ODT as formulations?
Thank you for the comment.
In order to address the reviewer comment, we have made changes to the text: please see, lines 46 - 50.
2. Table 1 contains links on 29 APIs from literature. But how 256 API were obtained? If 145 unique records were retrieved from "the original database by Han et al. [16]" and about 29 were retrieved from literature, then how it can give 256 in total? How many unique samples were used for training? There are a lot of questions that need answer.
The database by Han et al. consisted of 145 unique data records coming from undisclosed number of articles, which consisted of 26 APIs (taking into account authors error with synonymous name, Acetaminophen = Paracetamol, API number is actually 25). It means that 1 API was the source of one or more data records (formulations), differing in composition, process parameters and disintegration time.
Our database was constructed in a similar manner. However, Han et al. did not provided references to the sources of publications. Therefore, we had to recognize the sources, then we have critically reviewed the data provided in the publications and we have included only the data which in our opinion was robust enough to produce concise database. During the review we have also corrected errors in the former database. In the end of the process of data scrapping we had 245 unique data records (not API’s but formulations) coming from 26 unique APIs.
The detailed information and source files of 10-fold cross validation splits (training-testing pairs) are available at the https://github.com/jszlek/ODT_database
In order to address the comments, we have made changes to the text: please see, lines 166 – 168, 213 – 216, 365.
2. Table 3 contains a mix of hyperparameters for various machine learning methods. For which method RMSE is provided? Is it average RMSE? Please, specify.
Thank you for the comment.
As stated in the table 3 caption, the RMSE, NRMSE and R2 values are provided for the average from 30 repetitions of AutoML procedure, during which an algorithm is developing models with all available methods. The resulting final models could be of any ML method. The aim of presenting the results in the table was to show robustness of the methodology. In order to
In order to address the reviewer comment, we have made changes in the text (please see lines 397 - 399) and the table 3 caption (please see lines 412 - 413).
3. Please explain physical meaning of selected features and their possible impact on the disintegration time of ODTs.
Thank you for your valuable note.
The authors agree that the physical meaning of the selected features and their possible impact on the ODTs’ disintegration time should be covered in the text. We have included description of them in the section ‘3.1. Database’ (please see lines 349 – 363) and in the section ‘3.2. Feature selection and final model development’ (please see lines 402 – 446, and Table 4). Unfortunately, despite an intensive search of literature data, no scientific articles regarding the direct relationship between selected molecular descriptors and the physico-chemical properties of molecules were found. Therefore, their possible impact on the disintegration time is discussed based on the model’s explanation (please see lines 460 - 563. Where our results were indirectly confirmed by another research it was discussed in more details in the next section (please see lines 584 – 638). Moreover, we have found the way of the presentation of the results and discussion more convenient to the reader, because it gradually increases the complexity and detail of the description.
"It is common to train a variety of models on very large data sets without setting a priori". The phrase looks unclear.
Thank you for the comment.
The changes were made in order to rephrase the sentence. Please see lines 126 – 127.
Figures need more readable text and axex caption.
Thank you for your valuable note.
When presenting the figures, our intention was to minimize the changes of the source figures created using Python libraries such as h2o or SHAP. Due to the above, we did not change the size and type of the fonts. Moreover, despite strenuous attempts, we have failed to make Word generate docx or pdf files with the intended figures’ resolution (300 dpi), even if no compression was used. Therefore, we included high resolution (300 dpi) figures and SVG source files to the original submission of the article (please see attached zip file to the original submission). As a proof of our concept, please find attached link to Figure 8 (Figure 9.pdf) produced with external software (high resolution PDF file), which in our opinion is balanced between readability of the plot itself and arresting the reader’s attention to the axes only. Of course, we will closely cooperate with the Editorial Office in order to supply the best possible solution to the readers.
Reviewer 2 Report
The Authors present a well-done study on the formulations of orally disintegrating tablets (ODT) and their influence on disintegration time. The Authors collected data from literature and developed a final machine learning approach based on a deep neural network model with four layers. The results appear good in terms of predictive performance. The interpretation of results, in terms of relevance of the variable and effect of the pharmacological ingredient appears soundable.
I would note few points to be assessed.
As the Authors note under Conclusions, the availability of data is a crucial point. It could be interesting to explore the possible results, or at least discuss the expectable results, with a different selection at level of database construction (as an example, by adding also records without complete quality attributes).
Although present in the abstract, the Authors should explicit th API acronym under Introduction.
Table 3: I expected to read in the first line the headers of the columns. If it is, the headers are unclear.
Figures: most of the figures present labels or number difficult to read, unless I zoom in on the images with loss of resolution. Larger fonts should be used, and resolution increased.
Author Response
Dear Editor and Reviewers,
We were pleased to have an opportunity to revise our manuscript entitled “Puzzle out machine learning model - explaining disintegration process in ODTs” (1653469). In a revised manuscript, we have carefully considered reviewers’ comments and suggestions. As instructed, we have attempted to succinctly explain changes made in reaction to all comments. We reply to each comment in point-by-point fashion. We have color coded revised manuscript as text. The responses to the concerns raised by reviewers are below and are color coded as follows: a) Comments from reviewers are shown as text; b) Our responses are shown as text; c) Changes in the manuscript according to reviewers’ suggestions are highlighted.
The reviewers’ comments were very helpful overall, and we are appreciative of such constructive feedback on our original submission. After addressing the issues raised, we feel the quality of the paper is much improved. We hope that the manuscript in its present form is suited for publication.
Sincerely,
On behalf of all authors,
Jakub Szlęk
Reviewer #2
As the Authors note under Conclusions, the availability of data is a crucial point. It could be interesting to explore the possible results, or at least discuss the expectable results, with a different selection at level of database construction (as an example, by adding also records without complete quality attributes).
Thank you for the comment. We have added a paragraph to the Conclusions section as the Reviewer suggested. Please see lines 649 – 659.
“A data-driven modeling paradigm, that current AI/ML is based on, demands both high quality and large quantity of data. The former ensures precision (high predictability), whereas the latter accounts for a scope of the developed models. Given the highly automated manner of contemporary AI/ML implementations, search of crucial variables and handling of missing data by e.g. data imputation became also a domain of AI/ML. Having that said, we presume that when the dataset could be extended both in number of cases and features, the resulting models would retain or improve their efficacy yet broaden their scope. The quantity of data could be also a factor improving handling of incomplete features, when remaining cases would provide means for data imputation. This is of course case-related, yet AI/ML works surprisingly well on the filling the holes in the data when provided with large number of cases to analyze.”
Although present in the abstract, the Authors should explicit th API acronym under Introduction.
Thank you for the comment.
As suggested by the reviewer the API acronym was given. Please see lines 43 – 44.
Table 3: I expected to read in the first line the headers of the columns. If it is, the headers are unclear.
Thank you for the comment.
As suggested by the reviewer the table 3 was transposed. Please see lines 409 – 415.
Figures: most of the figures present labels or number difficult to read, unless I zoom in on the images with loss of resolution. Larger fonts should be used, and resolution increased.
Thank you for your valuable note.
When presenting the figures, our intention was to minimize the changes of the source figures created using Python libraries such as h2o or SHAP. Due to the above, we did not change the size and type of the fonts. Moreover, despite strenuous attempts, we have failed to make Word generate docx or pdf files with the intended figures’ resolution (300 dpi), even if no compression was used. Therefore, we included high resolution (300 dpi) figures and SVG source files to the original submission of the article (please see attached zip file to the original submission). As a proof of our concept, please find attached Figure 8 (Figure 8.pdf) produced with external software (high resolution PDF file), which in our opinion is balanced between readability of the plot itself and arresting the reader’s attention to the axes only.
Round 2
Reviewer 1 Report
Dear authors,
thank you for the comments.
I have one minor comment regarding the reply on my previous comment #2:
Table 1 contains links on 29 APIs from literature. But how 256 API were obtained? If 145 unique records were retrieved from "the original database by Han et al. [16]" and about 29 were retrieved from literature, then how it can give 256 in total? How many unique samples were used for training? There are a lot of questions that need answer.
I've looked through the (new) text and I suggest to re-phrase the text with the number of the records of API and how many formulations were there in Han's database, and the number of records (API and formulations) collected by the authors disregarding number of articles processed. The number of articles processed can be given in another sentence and even paragraph. My proposition is made to make text clearer for a reader.
Author Response
Dear Reviewer,
We would like to thank you for your valuable time and effort in reviewing our manuscript. After addressing the issue raised, we feel the paper in its present form is suited for publication. The responses to the concern raised by reviewer are below and are color coded as follows: a) Comments from reviewers are shown as text; b) Our responses are shown as text; c) Changes in the manuscript according to reviewers’ suggestions are highlighted.
Reviewer #1
I have one minor comment regarding the reply on my previous comment #2:
Table 1 contains links on 29 APIs from literature. But how 256 API were obtained? If 145 unique records were retrieved from "the original database by Han et al. [16]" and about 29 were retrieved from literature, then how it can give 256 in total? How many unique samples were used for training? There are a lot of questions that need answer.
I've looked through the (new) text and I suggest to re-phrase the text with the number of the records of API and how many formulations were there in Han's database, and the number of records (API and formulations) collected by the authors disregarding number of articles processed. The number of articles processed can be given in another sentence and even paragraph. My proposition is made to make text clearer for a reader.
We agree that the text should be clearly written. Therefore, the changes were made in order to rephrase the sentences.
Please see lines 120 and 164 - 169.